# Structure-Based Identification of Natural Products as SARS-CoV-2 M^pro^ Antagonist from *Echinacea angustifolia* Using Computational Approaches

**DOI:** 10.3390/v13020305

**Published:** 2021-02-15

**Authors:** Shiv Bharadwaj, Sherif Aly El-Kafrawy, Thamir A. Alandijany, Leena Hussein Bajrai, Altaf Ahmad Shah, Amit Dubey, Amaresh Kumar Sahoo, Umesh Yadava, Mohammad Amjad Kamal, Esam Ibraheem Azhar, Sang Gu Kang, Vivek Dhar Dwivedi

**Affiliations:** 1Department of Biotechnology, Institute of Biotechnology, College of Life and Applied Sciences, Yeungnam University, 280 Daehak-Ro, Gyeongsan, Gyeongbuk 38541, Korea; shiv@ynu.ac.kr; 2Special Infectious Agents Unit, King Fahd Medical Research Center, King Abdulaziz University, 21589 Jeddah, Saudi Arabia; saelkfrawy@kau.edu.sa (S.A.E.-K.); talandijany@kau.edu.sa (T.A.A.); lbajrai@kau.edu.sa (L.H.B.); prof.ma.kamal@gmail.com (M.A.K.); 3Department of Medical Laboratory Technology, Faculty of Applied Medical Sciences, King Abdulaziz University, Jeddah 21589, Saudi Arabia; 4Biochemistry Department, Faculty of Sciences, King Abdulaziz University, Jeddah 21589, Saudi Arabia; 5Department of Biosciences, Integral University, Lucknow 226026, India; altaf.bioinfo321@gmail.com; 6Computational Chemistry and Drug Discovery Division, Quanta Calculus Pvt. Ltd., Kushinagar 274203, India; ameetbioinfo@gmail.com; 7Department of Applied Sciences, Indian Institute of Information Technology Allahabad, Allahabad 211015, Uttar Pradesh, India; asahoo@iiita.ac.in; 8Department of Physics, Deen Dayal Upadhyay Gorakhpur University, Gorakhpur 273009, India; u_yadava@yahoo.com; 9Enzymoics, 7 Peterlee Place, Novel Global Community Educational Foundation, Hebersham, NSW 2770, Australia; 10Centre for Bioinformatics, Computational and Systems Biology, Pathfinder Research and Training Foundation, Greater Noida 201308, India

**Keywords:** SARS-CoV-2, COVID-19, natural products, *Echinacea-angustifolia*, molecular dynamics simulation, Quercetagetin 7-glucoside

## Abstract

Coronavirus disease-19 (COVID-19) pandemic, caused by the novel SARS-CoV-2 virus, continues to be a global threat. The number of cases and deaths will remain escalating due to the lack of effective therapeutic agents. Several studies have established the importance of the viral main protease (M^pro^) in the replication of SARS-CoV-2 which makes it an attractive target for antiviral drug development, including pharmaceutical repurposing and other medicinal chemistry approaches. Identification of natural products with considerable inhibitory potential against SARS-CoV-2 could be beneficial as a rapid and potent alternative with drug-likeness by comparison to de novo antiviral drug discovery approaches. Thereof, we carried out the structure-based screening of natural products from *Echinacea-angustifolia*, commonly used to prevent cold and other microbial respiratory infections, targeting SARS-CoV-2 M^pro^. Four natural products namely, Echinacoside, Quercetagetin 7-glucoside, Levan N, Inulin from chicory, and 1,3-Dicaffeoylquinic acid, revealed significant docking energy (>−10 kcal/mol) in the SARS-CoV-2 M^pro^ catalytic pocket via substantial intermolecular contacts formation against co-crystallized ligand (<−4 kcal/mol). Furthermore, the docked poses of SARS-CoV-2 M^pro^ with selected natural products showed conformational stability through molecular dynamics. Exploring the end-point net binding energy exhibited substantial contribution of Coulomb and van der Waals interactions to the stability of respective docked conformations. These results advocated the natural products from *Echinacea angustifolia* for further experimental studies with an elevated probability to discover the potent SARS-CoV-2 M^pro^ antagonist with higher affinity and drug-likeness.

## 1. Introduction

Coronavirus disease 2019 (COVID-19) is caused by a novel positive-sense single-stranded RNA virus named severe acute respiratory syndrome coronavirus-2 (SARS-CoV-2). This virus has been classified under family Coronaviridae, subfamily *Coronaviridae* and order nidovirales [1]. More than 69.8 million cases have been reported globally and have resulted in the death of more than 1.4 million people as of December 2020 [2]. The disease is associated with highly variable symptoms ranging from mild ones like fever, cough, and sneezing to severe life-threatening complications (e.g., acute respiratory distress syndrome and multiple organ failure). The RNA genome sequence of SARS-CoV-2 is about 30 kb in size that encodes a large polyprotein. Later, the polyprotein is digested into 4 structural proteins, such as spike (S), envelope (E), membrane (M), and nucleocapsid (N) as well as 16 non-structured proteins (nsp1-nsp16), and 8 accessory proteins. Each SARS-CoV-2 protein has its unique and key functions in the viral replication cycle. The viral protein nsp5, also termed as 3C-like proteinase (3CL^pro^) or main protease (M^pro^), is a key for processing of polyprotein into the functional proteins [3]. Hence, it has been recognized as an important anti-SARS-CoV-2 target due to its essential role in viral replication. Indeed, targeting and inhibiting the activity of M^pro^ might block or impair viral replication [4]. Several crystallographic and nuclear magnetic resonance (NMR) structures of SARS-CoV-2 M^pro^ have been solved in complex with different small molecules [3,5], which is critical achievement in discovering new therapeutics against COVID-19. However, there is no specialized prophylaxis or treatment for this virus till date, therefore, it demands an urgent requirement for the screening of new potent and effective drugs against SARS-CoV-2 [6].

In the area of drug discovery research, medicinal plants are one of the most popular sources of effective natural lead molecules. They contain a variety of phytochemicals that can be used as a drug against different diseases and infections [7]. There are several reports on the antiviral activity of numerous plants against highly pathogenic viruses, including coronaviruses [8,9]. The plants of *Echinacea genus* are one among those plants, which are well known for their potential antiviral activity against pathogenic coronaviruses [10,11,12,13]. *Echinacea angustifolia* (*E. angustifolia*) is one of the popular medical plants belonging to this genus with medicinal value, including antimicrobial and anti-inflammatory activities [14,15]. *E. angustifolia* have been reported to exhibited broad-spectrum antibacterial activity for several Gram-positive and Gram-negative bacteria like *Escherichia coli*, *Staphylococcus aureus*, *Bacillus subtilis*, and *Staphylococcus epidermidis* [16]. Interestingly, the antiviral activity of *E. angustifolia* is well known against several viruses like herpes simplex virus (HSV), influenza virus (FV), and rhinovirus (RV) [17]. Hence, this study was designed to identify the natural products in *E. angustifolia* with considerable potential to treat SARS-CoV-2 infection via SARS-CoV-2 M^pro^ inhibition. In this context, structure-based drug discovery methods, including molecular docking simulation, molecular dynamics (MD) simulation, and end point binding free energy approach were used to assess the therapeutic potential of natural products in *E. angustifolia* against SARS-CoV-2 by targeting M^pro^ protein as depicted in Figure 1.

## 2. Methodology

### 2.1. Receptor and Ligands Collection

Three-dimensional (3D) X-ray crystal structure of SARS-CoV-2 M^pro^ (PDB ID: 5R82) [18] co-crystallized with 6-(ethylamino)pyridine-3-carbonitrile at 1.31 Å resolution was downloaded as receptor from the RCSB protein database (http://www.rcsb.org/pdb/home/home.do) (accessed on 10 January 2021) [19]. Besides, the natural products reported in *Echinacea angustifolia* were collected from the literature and their respective 2D or 3D structures were downloaded in ‘sdf’ format from the PubChem database (https://pubchem.ncbi.nlm.nih.gov) (accessed on 10 January 2021) [20].

### 2.2. Structure-Based Ligand Identification and Quantum Chemical Calculations

Initially, 2D and 3D structures of the collected natural products reported in *E. angustifolia* were prepared for the molecular docking using LigPrep module in the Schrödinger suite (Schrödinger Release 2019-2: LigPrep, Schrödinger, LLC, New York, NY, USA, 2019). Moreover, the crystal structure of SARS-CoV-2 M^pro^ was also refined for the molecular docking using PRIME and protein preparation wizards in Schrödinger suite (Schrödinger Release 2019.2: Prime, Schrödinger, LLC, New York, NY, USA, 2019). For the identification of natural products from *E. angustifolia* with the potential to block SARS-CoV-2 M^pro^, the active residues (His^41^, Ser^46^, Met^49^, Cys^145^, His^164^, Met^165^, Phe^181^, Asp^187^, Arg^188^, and Gln^189^) interacting with the co-crystalized ligand, i.e., 6-(ethylamino)pyridine-3-carbonitrile, were considered in the molecular docking method, as described in crystal structure of SARS-CoV-2 M^pro^ [18]. Next, molecular docking simulation was performed under default parameters using extra precision (XP) protocol of the GLIDE tool in Schrödinger suite (Schrödinger Release 2019.2: Glide, Schrödinger, LLC, New York, NY, USA, 2019). During this process, receptor was treated as a rigid entity while ligands were considered as flexible to attain the most feasible interactions with the active residues in the catalytic pocket of SARS-CoV-2 M^pro^. Additionally, polar interactions, Coulombic, hydrogen bond, hydrophobic contacts, van der Waals, metal binding, freezing rotatable bonds, water desolvation energy, binding affinity enriching interactions, and the penalty for buried polar groups were also considered in the Glide extra precision (XP) scoring approach [21]. Following, ligands exhibiting highest negative docking score in SARS-CoV-2 M^pro^ catalytic pocket were considered for further structure refinement using the quantum chemical calculation in GAUSSIAN-03 suite [22]. The selected natural products along with co-crystallized ligand as a reference compound, i.e., 6-(ethylamino)pyridine-3-carbonitrile, were treated with the density functional theory (DFT) [23] under hybrid functional Becke’s three-parameter and the Lee–Yang–Parr functional (B3LYP) [24,25] in conjunction with 6-31G(d,p) basis sets, as reported earlier [26]. The molecular geometries of each natural product as ligand were fully optimized without constraint and respective global minima for the potential energy surface of each ligand were calculated devoid of imaginary frequency modes. Later, the computed optimized geometries were further studied for the molecular properties using GAUSSIAN-03 suite [22].

### 2.3. Re-Docking Simulation and Pose Profiling

The DFT optimized geometries of the natural products were re-docked in the catalytic pocket of SARS-CoV-2 M^pro^ using extra precision (XP) protocol of the GLIDE module in Schrödinger suite, as discussed in the earlier section. Next, at least 10 poses were generated for each docked ligand with SARS-CoV-2 M^pro^ and binding pose with highest negative docking score corresponds to least root mean square deviation (RMSD) were collected for intermolecular interactions in the academic Schrödinger-Maestro v12.4 suite (Schrödinger Release 2020-2: Maestro, Schrödinger, LLC, New York, NY, USA, 2020). Molecular contacts formed between the ligands and active residues of SARS-CoV-2 M^pro^ were evaluated within 4 Å area around the ligand in terms of noncovalent interactions, viz. hydrophobic interactions, hydrogen bonding, π-π interactions, π-cation interactions, positive interactions, negative interactions, glycine interactions, and formation of salt bridges under default parameters. Likewise, optimized co-crystallized ligand, i.e., 6-(ethylamino)pyridine-3-carbonitrile, was also docked in the binding region of SARS-CoV-2 M^pro^ to validate the docking protocol and for comparative analysis against selected natural products. Later, both 3D and 2D docked poses of viral protease with natural products were rendered in the academic Schrödinger-Maestro v12.4 suite (Schrödinger Release 2020-2: Maestro, Schrödinger, LLC, New York, NY, USA, 2020).

### 2.4. Explicit Solvent Molecular Dynamics Simulations

The best-docked poses of SARS-CoV-2 M^pro^-natural products were considered for molecular dynamics (MD) simulation analysis to envisage the stability of the selected docked complexes and intermolecular interaction as function of 100 ns time under a Linux environment on an HP Z2 Microtower workstation by academic version of Desmond v5.6 module [27] through Schrödinger-Maestro v11.8 suite interface (Schrödinger Release 2018-4: Desmond Molecular Dynamics System, D. E. Shaw Research, New York, NY, USA, 2018. Maestro-Desmond Interoperability Tools, Schrödinger, New York, NY, USA, 2018). Each docked complex was initially refined using the Protein preparation wizard and later covered in an orthorhombic grid box (10 × 10 × 10 Å buffer) amended with TIP4P (transferable intermolecular potential 4 points) water bath. The explicit solvent was selected for the MD simulation to recover most of the solvation effects of real solvent, including a contribution from the entropic origin such as the hydrophobic effect. In addition, counter Na^+^ and Cl^−^ ions were amended to nullify the charge of the whole system while placed at a distance of 20 Å around the ligand with the aid of a system builder tool. Additionally, salt of 0.15 M concentration was amended into the system to mimic the physiological conditions. Following, initial minimization of the whole system was performed under default parameters. This was followed by 100 ns molecular dynamics (MD) simulation at 300 K with default parameters. Finally, MD simulation trajectories collected at every 10 ps were analyzed for each simulated complex using the simulation interaction diagram (SID) tool of Desmond v5.6 package in Schrödinger-Maestro v11.8 suite.

### 2.5. Post Molecular Dynamics Simulation

#### 2.5.1. Essential Dynamics

Essential dynamics, in terms of principal component analysis (PCA), assisted in the collection of concerted motions linked with the largest atomic vibrations that are essentially required for the protein function [28,29]. Typically, >90% of the exhibited total atomic fluctuations can be defined by ≈20% of the principal axes, i.e., the covariance matrix eigenvectors [29]. Essential dynamics analysis was conducted on the respective MD simulation trajectories to assemble the PCs using Bio3d package [30]. The extraction of PCs was conducted based on all the Cα atoms of 5000 snapshots extracted from 100 ns MD simulation trajectory and aligned to the docked conformation pose to nullify the root mean square deviation (RMSD) between corresponding residues of the protein conformations. All the computations were performed on each simulated complex MD trajectory with the Bio3d package [30] under the R program environment [31].

#### 2.5.2. Binding Free Energy Calculations

In this study, binding free energy calculation was conducted using molecular mechanics generalized Born surface area (MM/GBSA), which is comparatively popular and more accurate approach against scoring function of molecular docking as well as computationally less demanding with respect to alchemical free energy methods [32]. Hence, net binding energy calculation were conducted using Prime MM/GBSA module by MM/GBSA protocol, as described earlier [33,34]. Herein, protein-ligand complex generated following molecular docking and snapshots extracted from 100 ns MD simulation trajectory (without explicit TIP4P water molecules and ions) for each complex was considered for net binding free energy calculation. The Equations (1)–(3) represents the mathematical expression to express the end-point binding free energy and respective individual decomposed energy components:(1)$${\Delta G}_{\mathrm{Bind}}={\Delta G}_{\mathrm{Com}}-\left({\Delta G}_{\mathrm{Rec}\;}+\;{\Delta G}_{\mathrm{Lig}}\right)=\Delta H-T\Delta S\;\approx {\;\Delta E}_{\mathrm{MM}}+{\Delta G}_{\mathrm{sol}}-T\Delta S,$$
(2)$${\Delta E}_{\mathrm{MM}}={\Delta E}_{\mathrm{Int}}+{\Delta E}_{\mathrm{Ele}\;}+{\Delta E}_{\mathrm{vdW}},$$
(3)$${\Delta G}_{\mathrm{Sol}}={\Delta G}_{\mathrm{Pol}}+{\Delta E}_{\mathrm{Nonpol}\;}.$$

The net binding free energy (ΔG_Bind_) is defined as the sum of free energy difference between the protein-ligand complex (G_Com_) and the free-state of each protein and ligand (G_Rec_ + G_Lig_). As the second law of thermodynamics stated, total binding free energy (ΔG_Bind_) of the protein-ligand complex is the sum of the enthalpy part (ΔH) and the entropy part (−TΔS) of the complete system, as given in Equation (1). 

To note, available computational methods for entropy calculation for the protein-ligand complex, including varying from post-processing methodologies [35,36,37,38,39] to the simulation-synchronized approaches [40,41,42], essentially required hundreds of nanoseconds simulation trajectories to explicitly compute the system entropy and such methodologies are only tested on a small system containing a few hundreds of atoms. Previous studies have suggested that these computational methodologies are not accessible to calculate the entropy of a system composed of thousands of atoms such as the protein-ligand complex [43]. Thus, contribution of entropy in the present study to calculate the net binding energy for the protein-ligand complexes was dropped due to high computational cost and relatively low contribution in total binding free energy, as reported earlier [43]. Given these conditions, enthalpy of the system can be assigned equal to the total binding free energy of the protein-ligand complex; and hence, expressed as the sum of molecular mechanical energy (ΔE_MM_) and solvation free energy (ΔG_Sol_). Characteristically, ΔEMM is defined as the sum of intramolecular energy (ΔE_Int_, which is the addition of the bond, angle, bond, and dihedral energies), the electrostatic energy (ΔE_Ele_), and the van er Waals interactions (ΔE_vdW_). Likewise, ΔGSol is the addition of net polar energy (ΔG_Pol_) and non-polar (ΔE_Nonpol_) of the system. Thus, total binding free energy were calculated using the MM/GBSA approach for each complex under default parameters with OPLS-2005 force field on the last 10 ns intervals of each 100 ns MD simulation trajectories. 

## 3. Results and Discussion

### 3.1. Structure-Based Ligand Identification 

Molecular docking simulation is a widely used computational approach in drug discovery. It enables discovering the favorable ligands with ideal conformation in the binding pocket of the target proteins. In this context, the XP docking algorithm in Glide aids in the identification of ligands with high affinities in the binding pocket of the receptor by including water desolvation energy, hydrophobic interactions, generation of neutral–neutral single or interrelated hydrogen bonds in a hydrophobically restricted environment, and other five clusters of charged–charged hydrogen bond formation [21]. Herein, a total of 50 natural products reported in *E. angustifolia* were docked in the binding pocket of viral main protease (M^pro^) using the Glide XP protocol, yields ligands with binding affinities in the range of -12.93 to 0.0897 kcal/mol, at least docking RMSD values as shown in Appendix A. One of the main objectives of this study was to identify the ideal poses of the docked ligands based on their docking score which can block the target protein. Hence, top five natural products, i.e., Echinacoside, Quercetagetin 7-glucoside, Levan N, Inulin from chicory, and 1,3-Dicaffeoylquinic acid, docked with SARS-CoV-2 M^pro^ were considered for further intermolecular interaction analysis. 

Echinacoside (ECH), a natural phenylethanoid glycoside, has been investigated for numerous pharmacologically benefits possessing high antiviral activities (e.g., against vesicular stomatitis virus) [44], and limited immune activation properties [45]. Whereas Levan is characterized as a naturally occurring polymer of fructan and typified by β-(2,6) linkages, and commonly present with diverse degree of polymerization in many microorganisms and plants species [46,47]. Levan isolated from *Bacillus* sp. strains showed antiviral activity against pathogenic avian influenza HPAI, H5N1, and adenovirus type 40 [48]. Remarkably, both H5N1 and SARS-CoV are RNA viruses that cause severe viral pneumonia leading to ARDS [49]. Likewise, Inulin is also a fructose polymer and differentiate from the Levan based on β-(2,1) bonds linkages. Unlike Levan, Inulin is only known for indirect viral inhibition by promoting and regulating the immune system [50]. Of note, a fructan composed of terminal (21.0%) and 2,1-linked β-d-Fruf residues (65.3%) with 1,6-linked β-d-Glcp residues (13.7%) isolated from Welsh onion (*Allium fistulosum* L.) was documented for in vivo inhibitory effect on influenza A virus replication [51]. Furthermore, 1,3-Dicaffeoylquinic acid (Cynarin) isolated from *Inula viscosa* was reported for exhibiting strong antioxidant activities via direct scavenging of several free radicals [52] while 1,3-Dicaffeoylquinic acid extracted from the leaves of *Cynara cardunculus* L. (Asteraceae) exhibited the inhibition of HIV-1 replication in MT-2 cell culture at non-toxic concentrations [53]. Recently, screening of anti-influenza lead compounds also identified a 1,3-Dicaffeoylquinic acid as an inhibitor of viral RNA polymerase [54]. Based on these reported literatures, it was concluded that the selected natural products possess medical properties, including antiviral activities; and hence, can be processed for further analysis within the catalytic pocket of SARS-CoV-2 M^pro^. 

### 3.2. Quantum Chemical Calculations 

#### 3.2.1. Geometry Optimization 

To calculate the molecular properties of the ligand in computational chemistry, hybrid functional B3LYP with 6-31(d,p) basis is well established for the geometry optimization of organic compounds. In this study, initially geometry optimization was performed and then structural parameters, including bond angles, dihedral angles, and bond length, were studied for the selected natural products, i.e., Echinacoside, Quercetagetin 7-glucoside, Levan N, Inulin from chicory, and 1,3-Dicaffeoylquinic acid, from *E. angustifolia* as antagonist against SARS-CoV-2 M^pro^ along with co-crystallized ligand 6-(ethylamino)pyridine-3-carbonitrile as reference compound (Appendix A). Figure 2 shows the 2D structure and 3D optimized geometries labeled with atom number scheme for the selected natural products and reference compound, which were rendered using GaussView 3.0.8.

#### 3.2.2. Frontier Molecular Orbitals Analysis 

In chemical reactions, the interaction and overlapping of two reactants’ molecular orbitals results leads to the formation of two new molecular orbitals, i.e., one as bonding orbitals containing less bonding energy and another as antibonding with higher energy [55]. Thus, the energy difference in the highly occupied molecular orbital (HOMO) and lower unoccupied molecular orbitals (LUMO), viz. E_HOMO_-E_LUMO_, for a chemical species, can be used to calculate its kinetic stability, chemical reactivity, and hardness [56,57]. Interestingly, a decrease in the frontier molecular orbital energy gap also indicates the intermolecular charge transfer (ICT) from donors to acceptor atoms and assist to elucidate the bioactivity of the molecule. Thereof, E_HOMO_, E_LUMO_, and E_HOMO_-E_LUMO_ values were computed from optimized geometries of the selected natural products. Figure 3 exhibits the asymmetric frontier molecular orbitals, i.e., HOMO and LUMO, for the potential natural products, i.e., (a) Echinacoside, (b) Quercetagetin 7-glucoside, (c) Levan N, (d) Inulin from chicory, and (e) 1,3-Dicaffeoylquinic acid. Herein, red and green color distribution was observed around the electronegative and electropositive atoms, respectively, demonstrates the corresponding negative and positive phases of the molecular frontier orbital wave function in the selected natural products. Likewise, reference compound 6-(ethylamino) pyridine-3-carbonitrile studied for the distribution of frontier molecular orbitals indicate formation of asymmetric HOMO and LUMO over electronegative and electropositive atoms on the optimized geometry (Appendix A). Hence, the electronegative and positive atoms in the respective ligand structures were predicted for formation of donor and acceptor molecular contacts in the active pocket of viral protease during molecular docking simulation. Furthermore, computed energy gap for the selected natural products also indicates the considerable kinetic stability and low chemical reactivity against reference compound (Figure 3 and Appendix A). To note, Levan N (6.57 eV) and Inulin from chicory (5.98 eV) showed higher energy gap by comparison to other selected natural products, i.e., Echinacoside (4.04 eV), Quercetagetin 7-glucoside (3.98 eV), 1,3-Dicaffeoylquinic acid (4.02 eV), and reference compound 6-(ethylamino)pyridine-3-carbonitrile (5.77 eV), suggested their substantial bioactivity and potential as antagonists against SARS-CoV-2 M^pro^. Conclusively, these observations suggested the considerable unreactive chemical behavior and chemical stability for the selected natural products, i.e., Echinacoside, Quercetagetin 7-glucoside, Levan N, Inulin from chicory and 1,3-Dicaffeoylquinic acid, against reference compound 6-(ethylamino) pyridine-3-carbonitrile. 

### 3.3. Re-Docking and Intermolecular Interaction Analysis

The DFT optimized structures of the potential natural products were further analyzed by re-docking in the catalytic pocket of SARS-CoV-2 M^pro^ to monitor the intermolecular interactions that can contribute to the stability of the respective docked complexes. Thus, selected natural products and reference ligand, viz. 6-(ethylamino) pyridine-3-carbonitrile, were re-docked in SARS-CoV-2 M^pro^ catalytic pocket by XP docking protocol. Following, the docked poses with highest negative docking score were considered for intermolecular interaction analysis. (Table 1, Figure 4, Appendix A). 

In molecular docked complexes, non-covalent interaction, such as electrostatic interactions, van der Waals interactions, salt bridges, hydrogen bonding, and metal interactions, are known to play a key role in the formation and stability of the receptor-ligand complex [58,59]. Remarkably, hydrogen bonding was reported to mediate the ligand binding with the receptor and fundamentally contribute to the physiochemical properties of the molecules, which are essentially required for the drug development of lead compounds [60,61]. Thus, each docked complex of natural products with SARS-CoV-2 M^pro^ was studied for intermolecular interactions at 4 Å distance around the ligand with default parameters using the 2D interaction tool in Maestro-Schrödinger suite (Figure 4). 

Molecular docked complexes of all the natural products, i.e., Echinacoside, Quercetagetin 7-glucoside, Levan N, Inulin from chicory, and 1,3-Dicaffeoylquinic acid, with SARS-CoV-2 M^pro^ exhibited substantial binding affinity >10 kcal/mol against <4 kcal/mol binding energy for the reference compound, viz. 6-(ethylamino)pyridine-3-carbonitrile) (Table 1). Additionally, a minimum of six hydrogen bond formation were observed for the selected natural products with active residues in the catalytic pocket of SARS-CoV-2 M^pro^ by comparison to the reference compound (only one hydrogen bond formation with residue Arg^188^). Of note, only SARS-CoV-2 M^pro^-Quercetagetin 7-glucoside and SARS-CoV-2 M^pro^-1,3-Dicaffeoylquinic acid docked complexes exhibited π-cation stacking (His^41^ residue) and salt bridge (His^41^ residue) formation (Figure 4). Additionally, substantial intermolecular interactions, such as hydrophobic, polar, negative, positive, π-π stacking, π-cation stacking, salt bridge, and glycine interactions, were noted in SARS-CoV-2 M^pro^-natural compounds against reference compound (Table 1, Figure 4, Appendix A). Interestingly, the interacting atoms of the respective ligands were predicted for molecular contacts with the active residues of the viral protease via electron donor and acceptor molecular contacts supporting the predicted sites on the respective optimized ligand geometries by DFT calculations for intermolecular interactions (Figure 3). Together, all the selected natural compounds were concluded as potent antagonists of SARS-CoV-2 M^pro^ against reference compound 6-(ethylamino) pyridine-3-carbonitrile; this observation was also supported by significant intermolecular interaction profiles for the respective docked complexes. 

### 3.4. Explicit Solvent Molecular Dynamics Simulation Analysis

Molecular dynamics simulation used in the drug discovery pipeline to establish the stability of small molecules docked with receptors computed from molecular docking simulations. The screened natural products docked with SARS-CoV-2 M^pro^ were also evaluated for the complex stability and formation of intermolecular interactions against reference ligand with respect to 100 ns simulation interval. Initially, the ligands were observed for the steadiness in the binding pocket of the viral protease at the end of 100 ns MD simulation by comparison to the initial frame revealed considerable stability for all the natural products, except for SARS-CoV-2 M^pro^-1,3-Dicaffeoylquinic acid docked complex which showed displacement of ligand from the catalytic pocket (Figure 5). Moreover, in reference docked complex, i.e., SARS-CoV-2 M^pro^-6-(ethylamino) pyridine-3-carbonitrile, exhibited diffusion of ligand from the catalytic pocket into unknown cavity of the viral protease during 100 ns MD simulation against initial protein-receptor pose (Figure 5). Furthermore, each last pose extracted from respective 100 ns MD trajectory were analyzed for the intermolecular contacts formation between the ligand and essential residues in the catalytic pocket of SARS-CoV-2 M^pro^ (Appendix A). Remarkably, all the natural products docked with viral protease were noted for the formation of substantial molecular contacts with the active residues in the catalytic pocket of viral protease, except for the reference ligand (Appendix A). Of note, at least two hydrogen bonds and other intermolecular interactions between the active residues of viral protease and the selected natural products were observed, suggesting the inhibitory potential of selected natural compounds against catalytic activity of SARS-CoV-2 M^pro^ (Appendix A, Appendix A). To take further account on the respective docked complexes stability, root mean square deviation (RMSD), root mean square fluctuation (RMSF), and protein-ligand contacts maps were extracted from the respective 100 ns MD simulation trajectories. 

#### 3.4.1. RMSD and RMSF Analysis

To calculate the average displacement in the protein and docked ligands in the respective complex during simulation interval, RMSD value for protein structure (Cα, backbone, sidechain, and heavy) and protein fit ligand were computed in reference to initial frame of the MD simulation trajectory (Figure 6, Appendix A). Interestingly, all the protein structures in the docked complexes with selected natural products exhibited acceptable RMSD values (<2 Å) against reference ligand docked complex (<3 Å) till the end of 100 ns MD simulation (Figure 6, Appendix A). These observations suggested the stability of the viral protease in the docked state with selected natural products, supported by considerable RMSD values (<3 Å) of the backbone, sidechain, and heavy atoms in the SARS-CoV-2 M^pro^ structure docked with respective natural products (Appendix A). Likewise, the protein fit ligand also showed deviations in the initial 10–50 ns interval and then followed by considerable variations and equilibrium state, except higher deviations were noticed for 1,3-Dicaffeoylquinic acid (<18.5 Å) till at the end of 100 ns simulation (Figure 6). Remarkably, only Quercetagetin 7-glucoside was observed with considerable deviations and acceptable RMSD (<3.5 Å) during 100 ns MD simulation interval (Figure 6, Appendix A). However, the reference ligand, i.e., 6-(ethylamino)pyridine-3-carbonitrile, docked with SARS-CoV-2 M^pro^ exhibited higher RMSD (>50 Å) within 10 ns followed by deviations between 10–60 ns and a state of equilibrium (<30 Å) from 60 ns till end of the 100 ns simulation interval (Appendix A). The initial high deviation value indicated the diffusion of ligand from the receptor within time frame of 10 ns and later equilibrium state indicates the placement of ligand in the unknown cavity of the receptor as predicted in the last snapshot of respective 100 ns MD trajectory (Figure 5, Appendix A). Hence, no further analysis was considered for the reference docked complex, i.e., SARS-CoV-2 M^pro^-6-(ethylamino) pyridine-3-carbonitrile. Moreover, RMSD values calculated for docked protein with natural products were further favored by acceptable RMSF values (<2 Å), except in the N- and C-terminal of the protein structure (<3.5 Å) (Appendix A). These end terminal fluctuations in the protein structure during the simulation can be ignored, which are for away from the catalytic pocket of the viral protease. Besides, natural products as fit ligand on protein also exhibited considerable RMSF values for the selected natural products, except in the atoms contributing molecular contacts with the active residues in the catalytic pocket of viral protease during the MD simulation interval (Appendix A). Collectively, although all the natural products were observed for stable complex formation with SARS-CoV-2 M^pro^ but docked complex of SARS-CoV-2 M^pro^-Quercetagetin 7-glucoside demonstrated substantial stability against the other selected natural products and reference ligand. 

#### 3.4.2. Protein-Ligand Contact Mapping

In drug design, hydrogen bonding (backbone acceptor; backbone donor; side-chain acceptor; side-chain donor) plays a key function in the drug metabolism, adsorption, and drug specificity. Moreover, hydrophobic interactions (π-cation; π-π; and other, and non-specific interactions), ionic interactions or polar interactions, and water bridge-hydrogen-bonded protein-ligand interactions mediated by a water molecule, also contributes to the stability of the docked complex during the simulation. Hence, intermolecular contact formed between SARS-CoV-2 M^pro^ and docked natural products, i.e., were extracted from the respective MD trajectories using default parameters of the Desmond module (Figure 7). 

Remarkably, all the docked natural products in the active pocket of SARS-CoV-2 M^pro^ were observed substantial intermolecular contact formation with catalytic residues, i.e., His^41^ and Cys^145^, and other substrate binding residues (Figure 7). Interestingly, these molecular contacts were also recorded in the respective docked complexes (Table 1, Figure 4), suggested the stability of docked complexes during simulation interval. Furthermore, all the natural products exhibited considerable number of intermolecular contacts, including hydrogen bonding, water bridging, hydrophobic, polar, negative, and positive interactions, with the active residues in the catalytic pocket of SARS-CoV-2 M^pro^ during 30% of the total MD simulation interval, except for SARS-CoV-2 M^pro^-Echinacoside docked complex (Figure 8). However, no considerable interactions were noted for the reference complex as ligand moved out of the catalytic pocket during the initial phase of 100 ns MD simulation (Figure 5). Conclusively, analysis of the protein-ligand contact mapping suggested the considerable occupancy of selected natural products in the catalytic pocket of SARS-CoV-2 M^pro^. Hence, the natural products from *E. angustifolia* can be marked in the order, i.e., Quercetagetin 7-glucoside, Inulin from chicory, Levan N, 1,3-Dicaffeoylquinic acid, and Echinacoside, as potent inhibitor of SARS-CoV-2 M^pro^ based on the number of intermolecular contacts formation during 30 % of the total 100 ns MD simulation interval. 

### 3.5. Essential Dynamics Analysis 

Essential dynamics is a statistical approach that assist in the extraction of the correlated motions in the protein structure in a trajectory produced by an MD simulation [28,62]. Thus, each simulation trajectory of docked SARS-CoV-2 M^pro^ with potential natural products: (a) Echinacoside, (b) Quercetagetin 7-glucoside, (c) Levan N, (d) Inulin from chicory, and (e) 1,3-Dicaffeoylquinic acid, was used in the construction of covariance matrix after removal of the rotational motions. Following, the diagonalization of the matrix was generated using the Bio3D package which results in the production of a set of eigenvectors/eigenvalues (Figure 9). Herein, each eigenvector exhibits one single direction in a multidimensional space while the eigenvalue represents the amplitude of the motion along the eigenvector. Figure 9 shows all the complexes with a steep drop in eigen fraction corresponds to the early five eigen modes and represent ~60% displacement in the residual motion of protein; these results represent the considerable induced conformation fluctuations the protein structure because of docked natural products as ligand in the catalytic pocket of SARS-CoV-2 M^pro^. Interestingly, a successive elbow point at the 5th eigen value followed by no momentous variations in the eigen fraction demonstrated a state of convergence in the respective complexes (Figure 9). These observations suggested that a significant flexibility was produced in SARS-CoV-2 M^pro^ during the initial phase of 100 ns MD simulation which eventually diminished to attain a stable complex formation with the docked ligands. Moreover, a steady decrement in the amplitude of an eigen fraction further demonstrates an additional localized fluctuation in the protein structure to achieve a favorable conformation. Thus, these fluctuations recorded in each complex may be considered as a requirement for the stability of the respective docked complexes during MD simulation as function of time. 

Moreover, the first three eigen vectors extracted from MD simulation trajectory of each selected complex were plotted to monitor the residual displacement in each protein structure where a color gradient change from blue to red stands through white color represents the periodic jumps among the extracted conformational poses of the docked protein structure (Figure 9). Of note, a considerable compact and variation in cluster distribution was observed for the residues of SARS-CoV-2 M^pro^ during 100 ns simulation, except in SARS-CoV-2 M^pro^-1,3-Dicaffeoylquinic acid complex (Figure 9). Besides, SARS-CoV-2 M^pro^-Quercetagetin 7-glucoside and SARS-CoV-2 M^pro^-Inulin from chicory complexes exhibited most favorable converged conformations and limited variation against other docked complexes during MD simulation; suggesting the considerable complex stability as noted from the respective RSMD and RMSF values (Figure 6, Appendix A). Conclusively, the eventual decrease in correlated and compact motions in SARS-CoV-2 M^pro^ structure in the respective docked complex demonstrates the induction of rigidity and complex stability of the respective complexes, except in residual motion induced by the docked ligand. In conclusion, a correlated fluctuations motion of the viral protease in all the studied systems represents the rigidity and the stability of the respective docked complexes, except in the SARS-CoV-2 M^pro^-1,3-Dicaffeoylquinic acid complex, during 100 ns MD simulation. 

### 3.6. Binding Free Energy

Computational calculations for computing binding affinities have been suggested as rapid and the most cost-effective methods to predict the binding affinities by considering the dynamics nature of the protein-ligand interaction in end-point free binding energy calculations such as MD-based MMGBSA methods. Thus, free energy for the binding of selected ligands in the active pocket of SARS-CoV-2 M^pro^ (ΔG_Bind_) were calculated using the MMGBSA method [63], implemented in the Prime MM/GBSA module of Schrödinger Suite. Herein, average binding free energy for the extracted poses from the last 10 ns simulation interval of each 100 ns simulation trajectory(Figure 10), except for the reference compound which diffused from the binding pocket. Of note, all the selected docked complexes exhibited binding free energy > 30 kcal/mol, maximum and minimum of binding free energy values were substantially noted for Quercetagetin 7-glucoside (−65.82 ± 2.74 kcal/mol) and 1,3-Dicaffeoylquinic acid (−42.57 ± 8.99 kcal/mol), respectively. Moreover, individual energy components contributing to net binding free energy, i.e., ΔG_Bind Coulomb_, ΔG_Bind Covalent_, ΔG_Bind Hbond_, ΔG_Bind Lipo_, ΔG_Bind Packing_, ΔG_Bind Solv GB_, and ΔG_Bind vdW_ were also computed in the MM/GBSA method (Figure 10, Appendix A). Remarkably, ΔG_Bind Coulomb_ and ΔG_Bind vdW_ were logged as the main contributors to the binding free energy for each ligand binding with the viral protease while ΔG_Bind Covalent_ and ΔG_Bind Solv GB_ were noted for contribution in unfavorable energy, therefore, reducing the net binding free energy values for each docked complex. These results agreed with a recent MMGBSA results for SARS-CoV-2 M^pro^ with Food and Drug Administration (FDA) approved drugs, in which ΔG_Bind Coulomb_ and ΔG_Bind vdW_ showed maximum contribution in the stabilization of the docked complexes [64]. Collectively, the selected natural products were deduced with substantial stability in the active pocket of viral protease as predicted from molecular docking and MD simulation analysis. 

## 4. Conclusions

This study aimed to identify the potent anti-SARS-CoV-2 compounds from E. angustifolia via SARS-CoV-2 M^pro^ inhibition through computational approaches. A total of five natural products were selected, i.e., Echinacoside, Quercetagetin 7-glucoside, Levan N, Inulin from chicory, and 1,3-Dicaffeoylquinic acid, based on substantial binding energy with catalytic pocket of SARS-CoV-2 M^pro^. Eventually, the selected natural products were noted with considerable bioactivity and formation of strong molecular contacts with the conserved residues in the catalytic pocket of the SARS-CoV-2 M^pro^, supported by molecular dynamics simulation and post simulation analysis. Based on the computed binding affinities, the selected compounds exhibited a potential inhibitory activity against the SARS-CoV-2 M^pro^ and can be used as potent antivirals against SARS-CoV-2 infection. This study provided an important step in the exploration of natural products from medical herbs for structure-based design of anti-SARS-CoV-2 drugs.

## Figures and Tables

**Figure 1 viruses-13-00305-f001:**
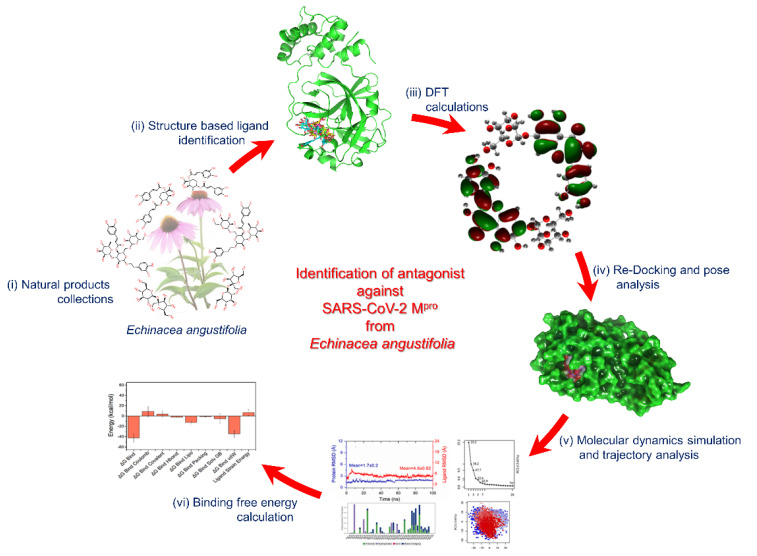
Computational assessment of natural products in *E. angustifolia* against SARS-CoV-2 M^pro^. Herein, (**i**) 3D structures of natural products reported in *E. angustifolia* were retrieved from the PubChem database, (**ii**) these natural products were then screened into the active region of SARS-CoV-2 M^pro^ by XP docking, (**iii**) natural products with highest negative docking energy were collected and treated by DFT method for respective geometry optimization to calculate other molecular properties, (**iv**) these optimized geometries of natural products were re-docked with SARS-CoV-2 M^pro^ and studied for pose binding and molecular contacts formation, (**v**) best docked poses were further studied for stability and protein ligand contact formation as function of 100 ns interval, and (**vi**) frames were extracted from respective MD trajectories and used in binding free energy calculations.

**Figure 2 viruses-13-00305-f002:**
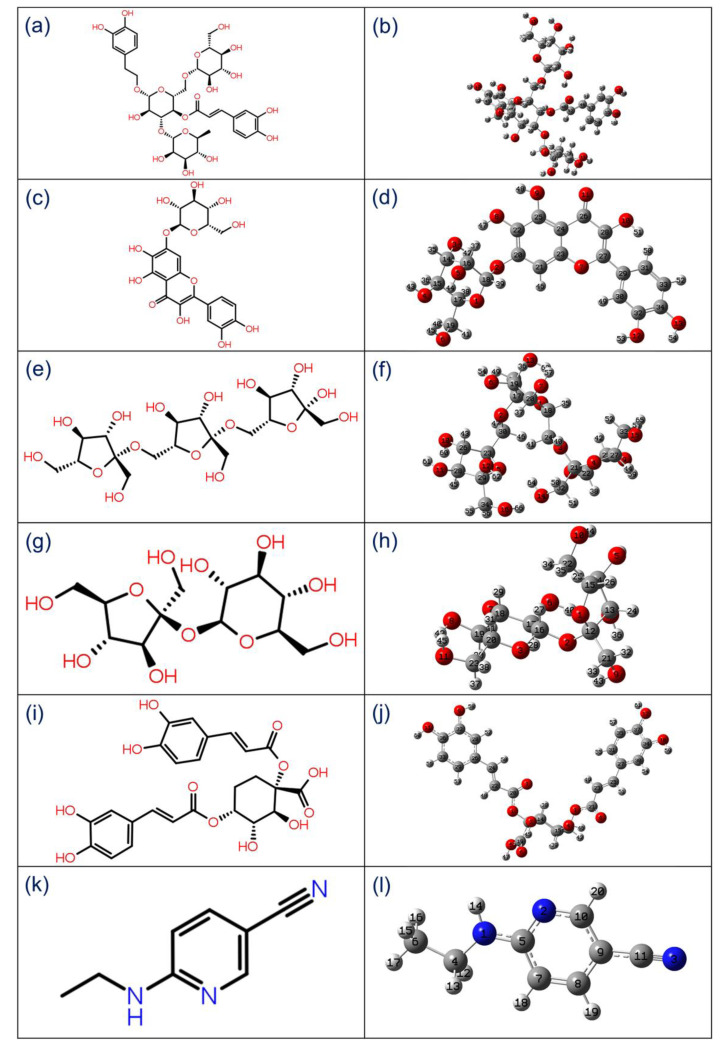
2D structures and 3D optimized molecular geometries of natural products, viz. (**a**,**b**) Echinacoside, (**c**,**d**) Quercetagetin 7-glucoside, (**e**,**f**) Levan N, (**g**,**h**) Inulin from chicory, (**i**,**j**) 1,3-Dicaffeoylquinic acid, and (**k**,**l**) 6-(ethylamino) pyridine-3-carbonitrile as reference compound, were calculated using DFT/B3LYP/6-31G(d,p) level.

**Figure 3 viruses-13-00305-f003:**
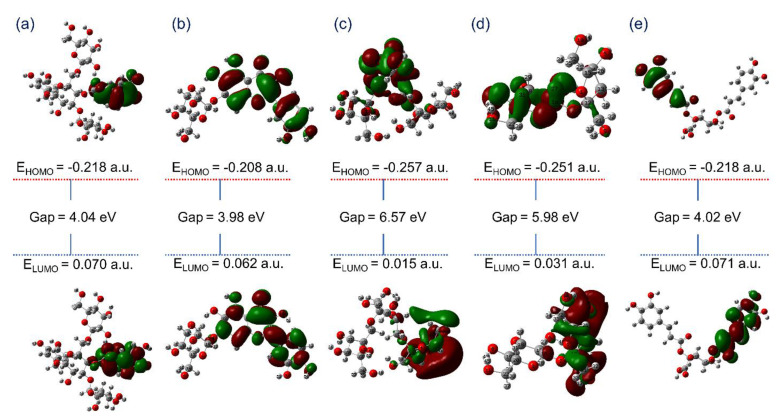
Molecular orbitals, i.e., highly occupied molecular orbital (HOMO) and lower unoccupied molecular orbitals (LUMO) of the optimized geometries of selected natural compounds, viz. (**a**) Echinacoside, (**b**) Quercetagetin 7-glucoside, (**c**) Levan N, (**d**) Inulin from chicory, and (**e**) 1,3-Dicaffeoylquinic acid, calculated using theoretical model B3LYP/6-31G** method.

**Figure 4 viruses-13-00305-f004:**
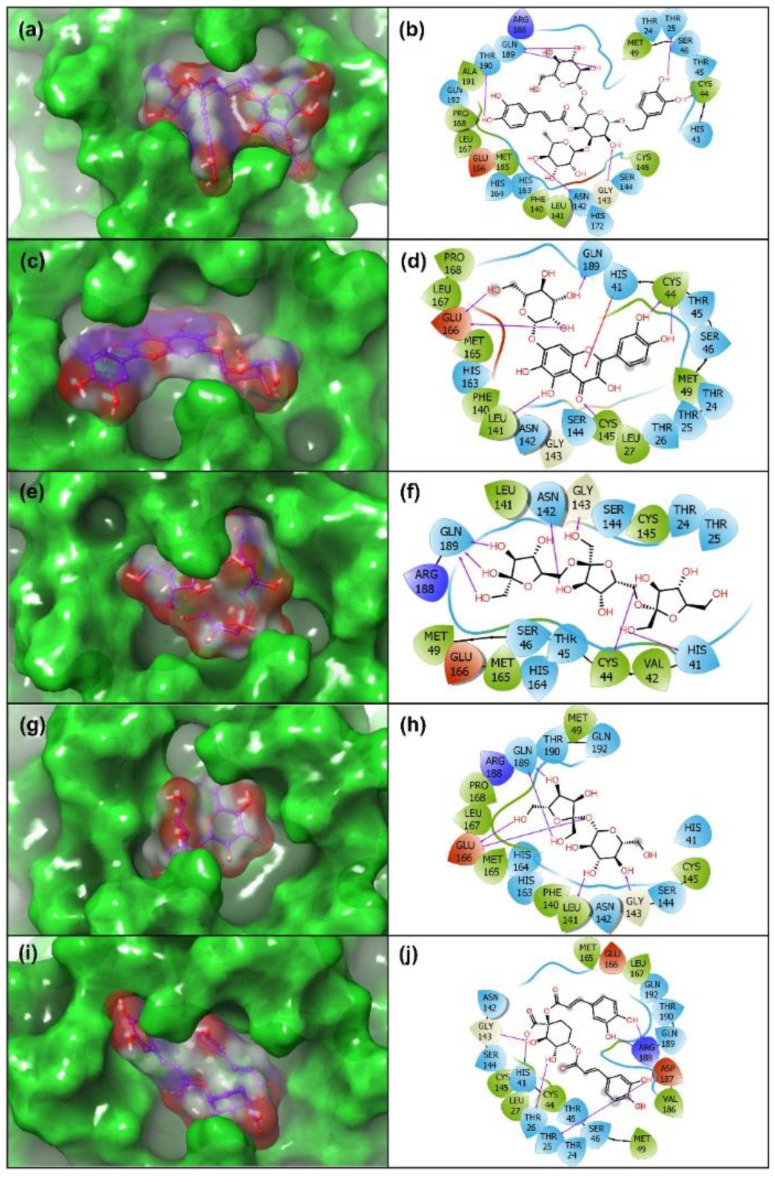
3D and 2D interaction maps for the docked poses of SARS-CoV-2 M^pro^ with potent natural products, i.e., (**a**,**b**) Echinacoside, (**c**,**d**) Quercetagetin 7-glucoside, (**e**,**f**) Levan N, (**g**,**h**) Inulin from chicory, and (**i**,**j**) 1,3-Dicaffeoylquinic acid. In 3D interaction poses, the docked ligand surface was generated based on partial charge while protein surface was rendered based on secondary structure. In 2D maps, hydrogen bond formation (pink arrows), hydrophobic (green), polar (blue), red (negative), violet (positive), glycine (grey), π-π stacking (green line), π-cation stacking (red line), and salt bridge (red-blue line), interactions are logged for docked complexes of SARS-CoV-2 M^pro^ with selected natural products.

**Figure 5 viruses-13-00305-f005:**
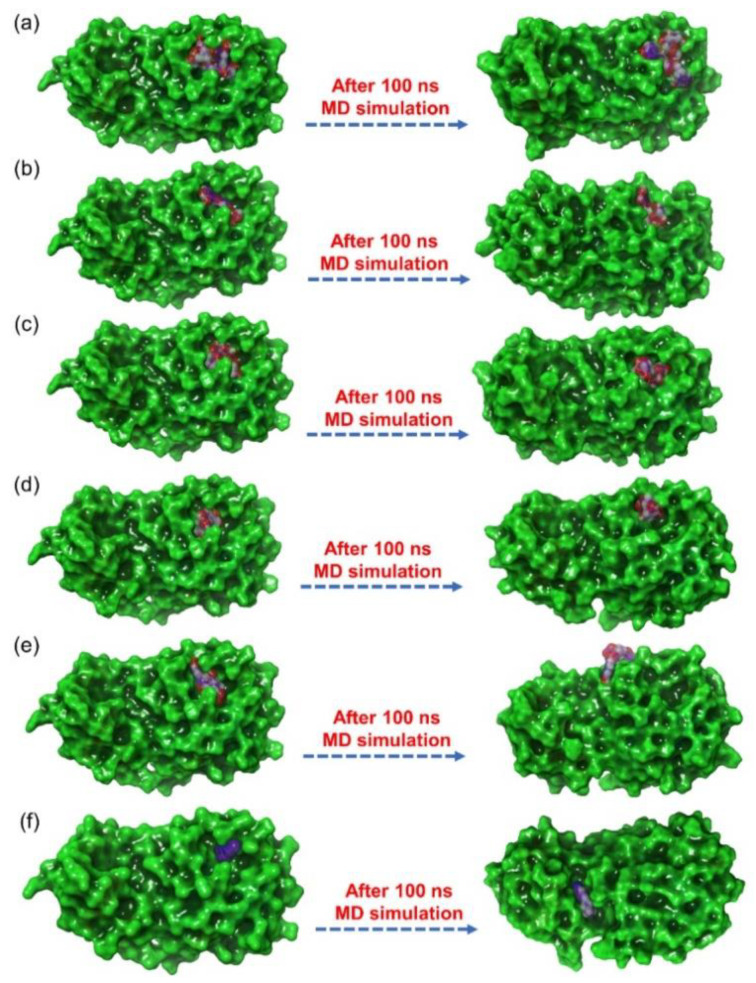
3D docked poses of SARS-CoV-2 M^pro^-natural products, i.e., (**a**) Echinacoside, (**b**) Quercetagetin 7-glucoside, (**c**) Levan N, (**d**) Inulin from chicory, and (**e**) 1,3-Dicaffeoylquinic acid, and (**f**) 6-(ethylamino) pyridine-3-carbonitrile, exhibiting transition of docked poses through 100 ns MD simulation.

**Figure 6 viruses-13-00305-f006:**
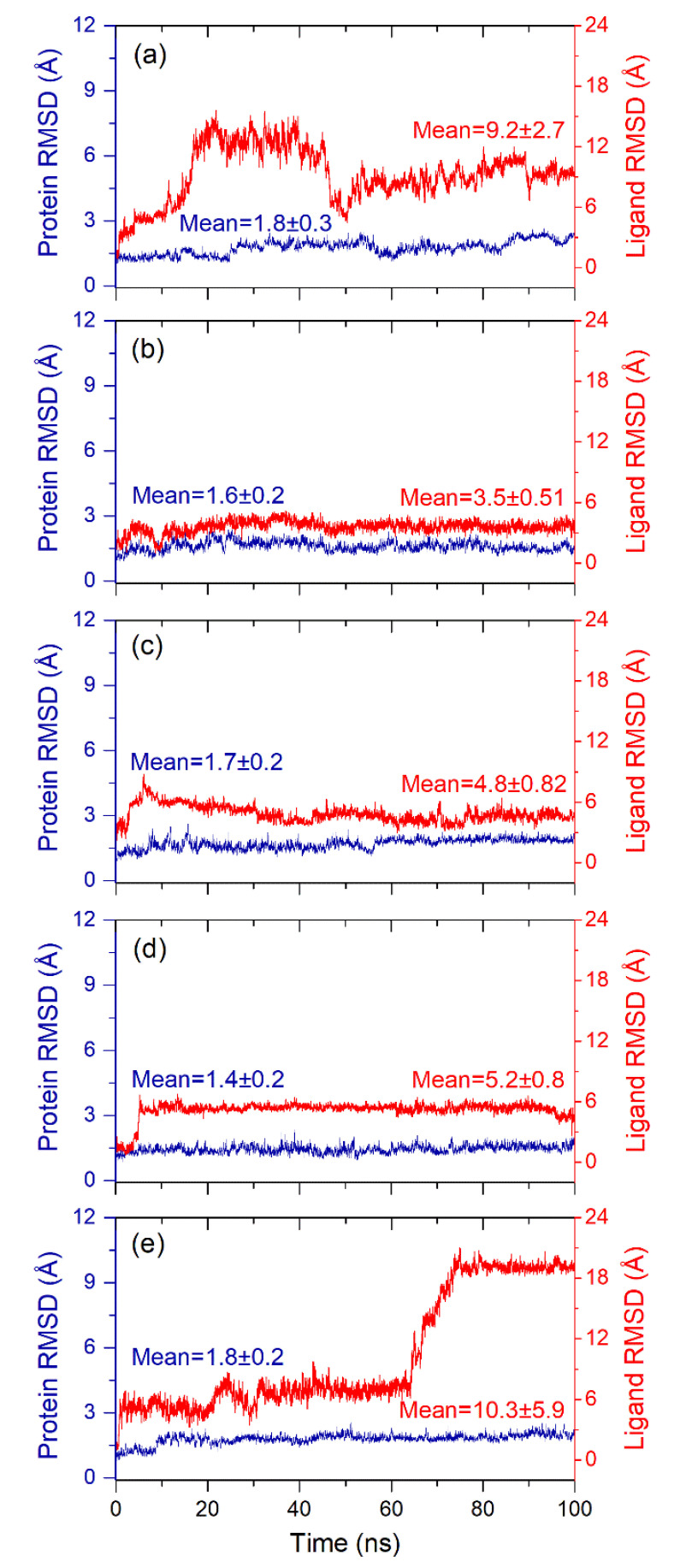
Root mean square deviation (RMSD) values plotted for alpha carbon atoms (blue curves) of SARS-CoV-2 M^pro^ and natural products (red curves), i.e., (**a**) Echinacoside, (**b**) Quercetagetin 7-glucoside, (**c**) Levan N, (**d**) Inulin from chicory, and (**e**) 1,3-Dicaffeoylquinic acid, were extracted from 100 ns MD simulation interval of respective docked complexes.

**Figure 7 viruses-13-00305-f007:**
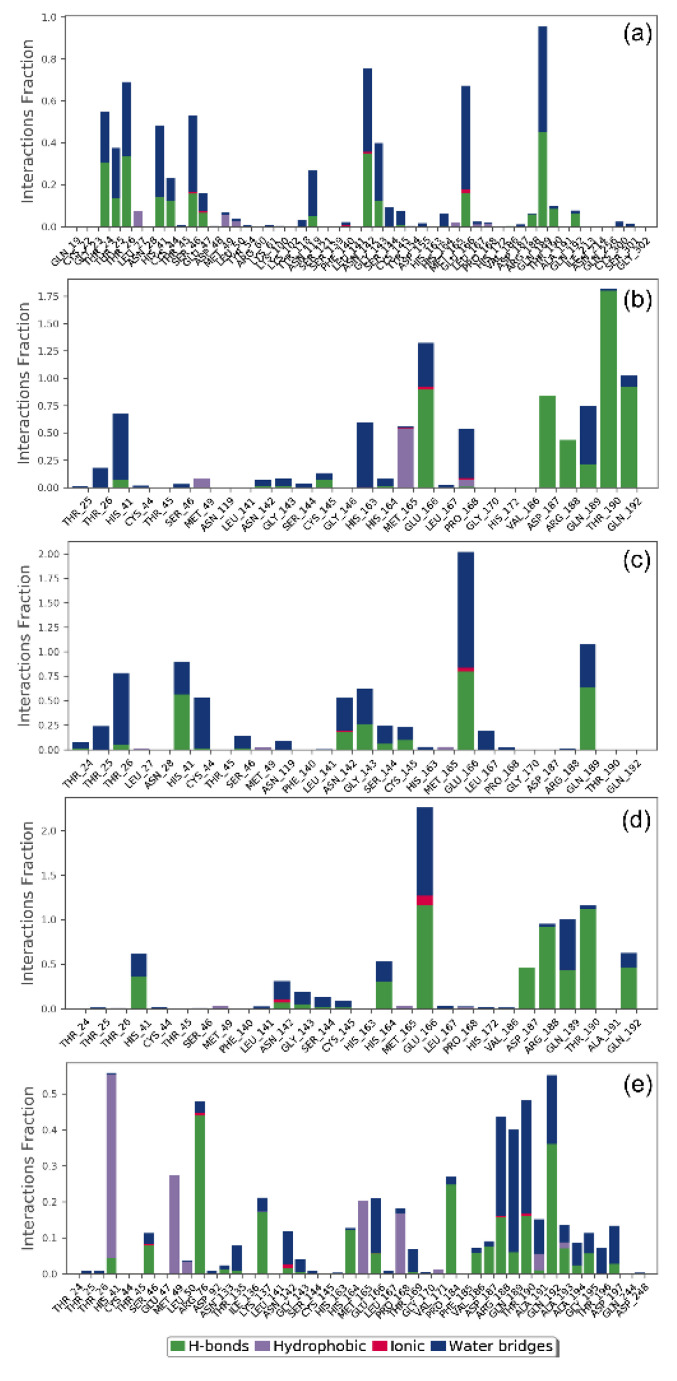
Protein-ligand interactions contact profiling for SARS-CoV-2 M^pro^ docked with potential natural products, viz. (**a**) Echinacoside, (**b**) Quercetagetin 7-glucoside, (**c**) Levan N, (**d**) Inulin from chicory, and (**e**) 1,3-Dicaffeoylquinica acid, computed from 100 ns MD simulation trajectories.

**Figure 8 viruses-13-00305-f008:**
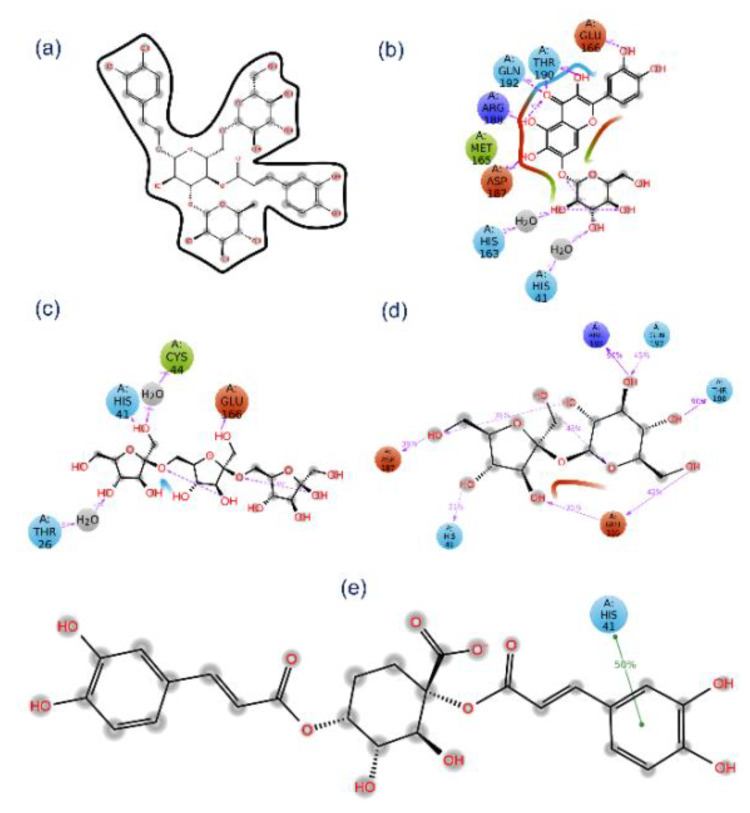
2D interaction diagrams for docked SARS-CoV-2 M^pro^-potential natural products, i.e., (**a**) Echinacoside, (**b**) Quercetagetin 7-glucoside, (**c**) Levan N, (**d**) Inulin from chicory, and (**e**) 1,3-Dicaffeoylquinic acid, are generated at 30% of the total 100 ns MD simulation interaction interval.

**Figure 9 viruses-13-00305-f009:**
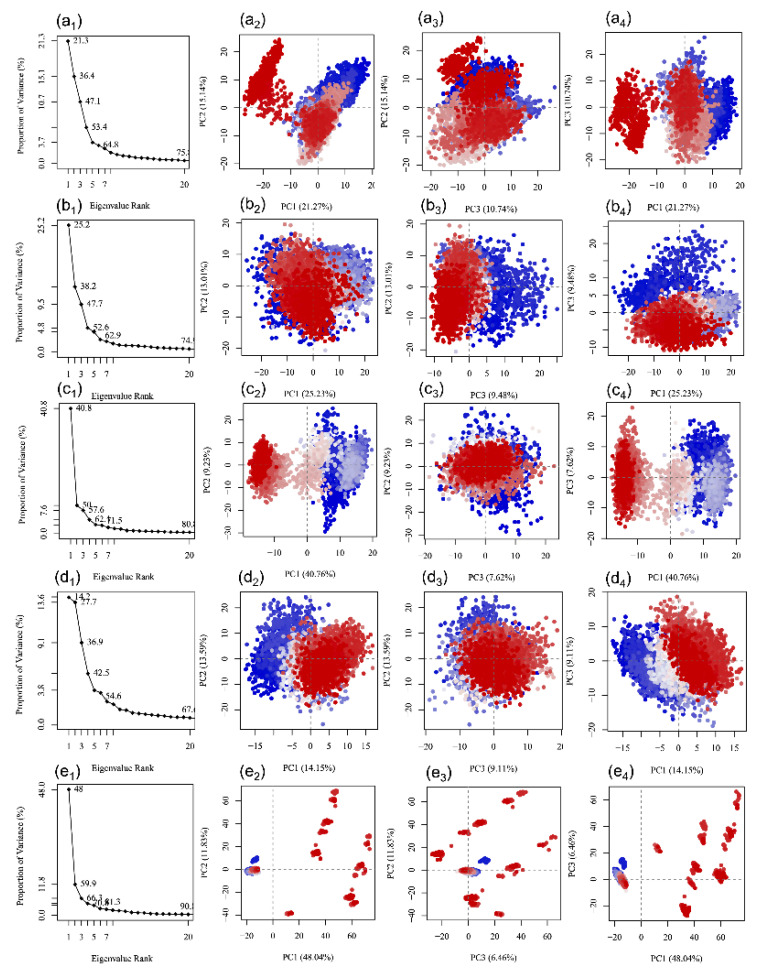
Principal component analysis for SARS-CoV-2 docked complexes with natural products, i.e., (**a**) Echinacoside, (**b**) Quercetagetin 7-glucoside, (**c**) Levan N, (**d**) Inulin from chicory, and (**e**) 1,3-Dicaffeoylquinic acid. The change from blue to red through white color in respective scatter plots shows the periodic jumps in the different conformations of the viral protease during 100 ns MD simulation intervals.

**Figure 10 viruses-13-00305-f010:**
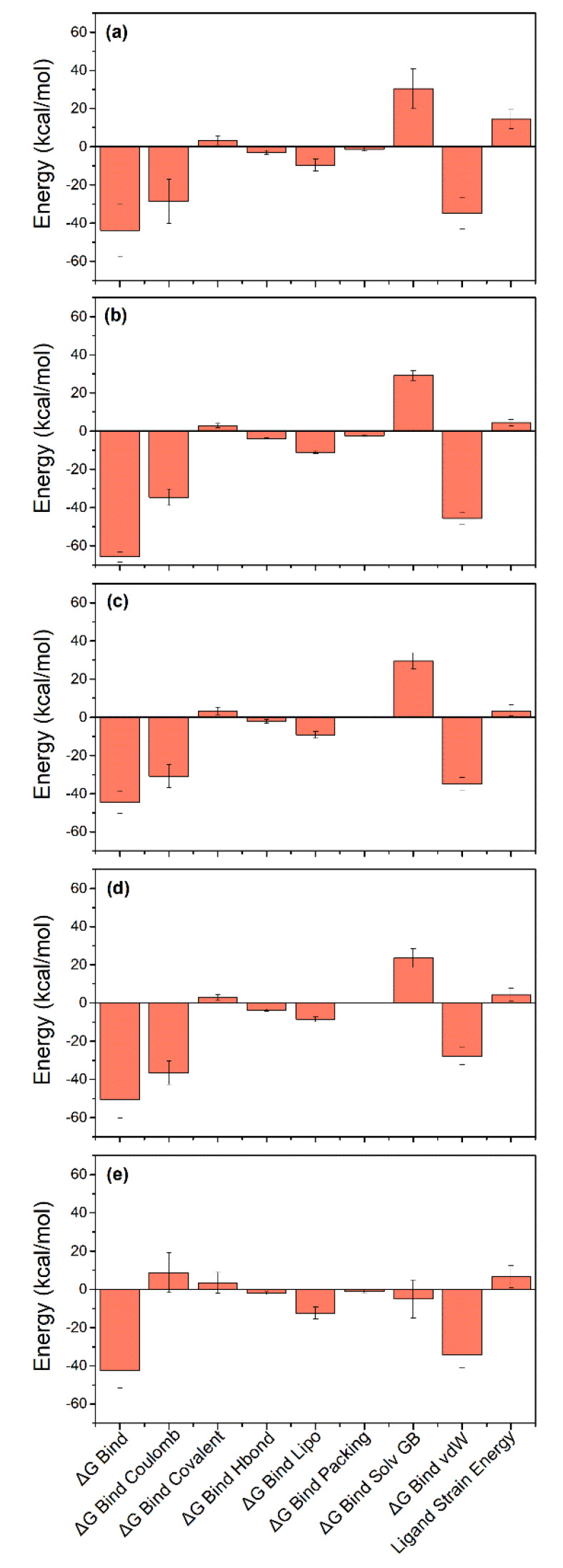
Average binding free energy and their dissociation energy components calculated for the extracted poses of SARS-CoV-2 M^pro^ with potent natural products, i.e., (**a**) Echinacoside, (**b**) Quercetagetin 7-glucoside, (**c**) Levan N, (**d**) Inulin from chicory, and (**e**) 1,3-Dicaffeoylquinic acid, from respective 100 ns MD trajectories.

**Table 1 viruses-13-00305-t001:** Intermolecular interactions observed for the best poses of docked natural products against reference compounds in the catalytic pocket of SARS-CoV-2 M^pro^ against the reference ligand.

S. no.	Compound	Docking Score (kcal/mol)	H-Bond	π-π/ * π- Cation Stacking	Hydrophobic	Polar	Negative	Positive	Glycine/ * Salt Bridge
1	Echinacoside	−14.17	Thr^25^, Cys^44^, Asn^142^, Gly^143^, Gln^189^, Thr^190^	-	Leu^27^, Val^42^, Cys^44^, Met^49^, Phe^140^, Leu^141^, Cys^145^, Met^165^, Leu^167^, Pro^168^, Ala^191^	Thr^24^, Thr^25^, His^41^, Thr^45^, Ser^46^, Asn^142^, Ser^144^, His^163^, His^164^, His^172^, Gln^189^, Thr^190^, Gln^192^	Glu^166^	Arg^188^	Gly^143^
2	Quercetagetin 7-Glucoside	−15.20	Cys^44^(2), Leu^141^, Cys^145^, Glu^166^(2), Gln^189^	* His^41^	Leu^27^, Cys^44^, Met^49^, Phe^140^, Leu^141^, Cys^145^, Met^165^, Leu^167^, Pro^168^	Thr^24^, Thr^25^, Thr^26^, His^41^, Thr^45^, Ser^46,^ Asn^142^, Ser^144^, His^163^, Gln^189^	Glu^166^	-	Gly^143^
3	Levan N	−12.92	His^41^, Cys^44^ Asn^142^, Gly^143^, Gln^189^(3)	-	Val^42^, Cys^44^, Met^49^, Leu^141^, Cys^145^, Met^165^	Thr^24^, Thr^25^, His^41^, Thr^45^, Ser^46^, Asn^142^, Ser^144^, His^164^, Gln^189^	Glu^166^	Arg^188^	Gly^143^
4	Inulin From Chicory	−11.72	Leu^141^, Gly^143^, Glu^166^ (2), Gln^189^(2)	-	Met^49^, Phe^140^, Leu^141^, Cys^145^, Met^165^, Leu^167^, Pro^168^	His^41^, Asn^142^, Ser^144^, His^163^, His^164^, Gln^189^, Thr^190^, Gln^192^	Glu^166^	Arg^188^	Gly^143^
5	1,3-Dicaffeoylquinic Acid	−10.01	Thr^26^,Thr^25^, Gly^143^, Arg^188^ (2)	-	Leu^27^, Cys^44^, Met^49^, Cys^145^, Met^165^, Leu^167^, Val^186^	Thr^24^, Thr^25^, Thr^26^, His^41^, Thr^45^, Ser^46,^ Asn^142^, Ser^144^, Gln^189^, Thr^190^, Gln^192^	Glu^166^, Asp^187^	Arg^188^	Gly^143^/* His^41^
6	6-(ethylamino)pyridine-3-carbonitrile	−3.57	Arg^188^	-	Met^49^, Cys^145^, Met^165^, Leu^167^, Pro^168^	His^41^, His^164^, Gln^189^, Thr^190^, Gln^192^	Glu^166^, Asp^187^	Arg^188^	Gly^143^

## Data Availability

Data are available in the article and in the Appendix A.

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
