# Peer review of "Structure-Based Identification of Natural Products as SARS-CoV-2 Mpro Antagonist from Echinacea angustifolia Using Computational Approaches"

_viruses, 2021, doi:10.3390/v13020305_

Round 1

Reviewer 1 Report

This original article is well written, and interesting. The authors claim that this study provided an important step in the exploration of natural products from medical herbs for structure-based design of anti-SARS-CoV-2 drugs via inhibiting the viral Mpro by utilizing computational simulations. Infact they well demonstrated that five natural compounds were selected on the bases of the binding energy with binding pocket of SARS-CoV-2 Mpro, with considerable bioactivity and formation of strong molecular contacts with the conserved residues in the catalytic pocket of the Mpro, supported by molecular dynamics simulation and post simulation analysis.
We have tested many molecules identified via computational simulation but almost all of them not working when tested in “vitro” experiment.
I strongly suggest to add some data about their effect "in vitro" on VERO E6 cells if available or at least on the purified Mpro enzyme to validate the computational results. Eventually the authors could find collaborations with a team available to screen these natural compounds and this would be a real advantage to validate the computational analysis

Author Response

Reviewer 1:

This original article is well written, and interesting. The authors claim that this study provided an important step in the exploration of natural products from medical herbs for structure-based design of anti-SARS-CoV-2 drugs via inhibiting the viral Mpro by utilizing computational simulations. Infact they well demonstrated that five natural compounds were selected on the bases of the binding energy with binding pocket of SARS-CoV-2 Mpro, with considerable bioactivity and formation of strong molecular contacts with the conserved residues in the catalytic pocket of the Mpro, supported by molecular dynamics simulation and post simulation analysis. We have tested many molecules identified via computational simulation but almost all of them not working when tested in “vitro” experiment.

Response:

We thank the reviewer for this compliment and hope that our in vitro evaluation for the studied compounds will be a success.  

I strongly suggest to add some data about their effect "in vitro" on VERO E6 cells if available or at least on the purified Mpro enzyme to validate the computational results. Eventually the authors could find collaborations with a team available to screen these natural compounds and this would be a real advantage to validate the computational analysis

Response:

We agree with the reviewer’s comments that an in vitro study is needed for validation of the computational study. But this is out of the scope of this manuscript which is solely focusing on the computational investigation of the activity for these compounds.

Reviewer 2 Report

The authors present a very thorough and high quality structure-based screening of natural products from Echinacea-angustifolia, against the SARS-CoV-2 Mpro protease catalytic pocket. The target protease is an excellent choice as a viral molecular target to discover new drugs. The study is timely to find new anti-viral drugs based on natural products, especially against CoV-19. The choice of Echinacea-angustifolia plant derived compounds has an ethnobotanical basis against respiratory ailments, therefore it is a good source for further study.

The computational work is excellent and professionally done, I have no criticism.

The manuscript would have benefited from “wet” validation experiments against the virus in in- vitro infected cells models. If proven effective in vitro, in vivo experiments would be welcomed.

Author Response

Reviewer 2:

The authors present a very thorough and high quality structure-based screening of natural products from Echinacea-angustifolia, against the SARS-CoV-2 Mpro protease catalytic pocket. The target protease is an excellent choice as a viral molecular target to discover new drugs. The study is timely to find new anti-viral drugs based on natural products, especially against CoV-19. The choice of Echinacea-angustifolia plant derived compounds has an ethnobotanical basis against respiratory ailments, therefore it is a good source for further study.

The computational work is excellent and professionally done, I have no criticism.

Response:

We thank the reviewer for this compliment.  

The manuscript would have benefited from “wet” validation experiments against the virus in in- vitro infected cells models. If proven effective in vitro, in vivo experiments would be welcomed.

Response:

We agree with the reviewer’s comments that an in vitro study is needed for validation of the computational study. But this is out of the scope of this manuscript which is solely focusing on the computational investigation of the activity for these compounds.

Round 2

Reviewer 1 Report

Agree